# Mutational Landscape of Bladder Cancer in Mexican Patients: *KMT2D* Mutations and chr11q15.5 Amplifications Are Associated with Muscle Invasion

**DOI:** 10.3390/ijms24021092

**Published:** 2023-01-06

**Authors:** María D. Pérez-Montiel, Dennis Cerrato-Izaguirre, Yesennia Sánchez-Pérez, Jose Diaz-Chavez, Carlo César Cortés-González, Jairo A. Rubio, Miguel A. Jiménez-Ríos, Luis A. Herrera, Anna Scavuzzo, Abelardo Meneses-García, Ricardo Hernández-Martínez, Felipe Vaca-Paniagua, Andrea Ramírez, Alicia Orozco, David Cantú-de-León, Diddier Prada

**Affiliations:** 1Departamento de Patología, Instituto Nacional de Cancerología (INCan), Mexico City 14080, Mexico; 2Departamento de Farmacología, Centro de Investigación y de Estudios Avanzados del I.P.N. (CINVESTAV), Avenida Instituto Politécnico Nacional No. 2508, Mexico City 07360, Mexico; 3Subdirección de Investigación Básica, Instituto Nacional de Cancerología (INCan), San Fernando No. 22, Tlalpan, Mexico City 14080, Mexico; 4Departamento de Urología, Instituto Nacional de Cancerología (INCan), Mexico City 14080, Mexico; 5Dirección General, Instituto Nacional de Medicina Genómica, Mexico City 14610, Mexico; 6Dirección GENERAL, Instituto Nacional de Cancerología (INCan), Mexico City 14080, Mexico; 7Laboratorio Nacional en Salud, Diagnóstico Molecular y Efecto Ambiental en Enfermedades Crónico-Degenerativas, Facultad de Estudios Superiores Iztacala, Tlalnepantla 54090, Mexico; 8Unidad de Apoyo Molecular a la Investigación Clínica, Instituto Nacional de Cancerología (INCan), San Fernando No. 22, Tlalpan, Mexico City 14080, Mexico; 9Dirección de Investigación, Instituto Nacional de Cancerología (INCan), Mexico City 14080, Mexico; 10Department of Environmental Health Sciences, Mailman School of Public Health, New York, NY 10032, USA

**Keywords:** bladder cancer, Hispanics, Mexican population, mutations, cancer genomics, non-muscle invasive bladder cancer, muscle-invasive bladder cancer

## Abstract

Bladder cancer (BC) is the most common neoplasm of the urinary tract, which originates in the epithelium that covers the inner surface of the bladder. The molecular BC profile has led to the development of different classifications of non-muscle invasive bladder cancer (NMIBC) and muscle-invasive bladder cancer (MIBC). However, the genomic BC landscape profile of the Mexican population, including NMIBC and MIBC, is unknown. In this study, we aimed to identify somatic single nucleotide variants (SNVs) and copy number variations (CNVs) in Mexican patients with BC and their associations with clinical and pathological characteristics. We retrospectively evaluated 37 patients treated between 2012 and 2021 at the National Cancer Institute—Mexico (INCan). DNA samples were obtained from paraffin-embedded tumor tissues and exome sequenced. Strelka2 and Lancet packages were used to identify SNVs and insertions or deletions. FACETS was used to determine CNVs. We found a high frequency of mutations in *TP53* and *KMT2D*, gains in 11q15.5 and 19p13.11-q12, and losses in 7q11.23. *STAG2* mutations and 1q11.23 deletions were also associated with NMIBC and low histologic grade.

## 1. Introduction

Bladder cancer (BC) is the most common neoplasm of the urinary tract. It generally originates from the epithelium that covers the inner surface of the bladder [1]. BC is the tenth most common cancer type, with an estimated 570,000 new cases worldwide in 2020 [2]. Smoking is the leading risk factor associated with BC incidence [3]. Within the total incidence rate, men tend to be diagnosed with bladder cancer 3 to 4 times more frequently than women. In addition, Non-Hispanic white patients are the most frequently affected, followed by Non-Hispanic blacks and Hispanics [4].

BC arises in two different pathways depending on the origin site. Non-muscular invading bladder cancer (NMIBC) or superficial BC represents about 60% of all BC cases. NMIBC is characterized by the loss of heterozygosity in chromosome 9 and mutations in *FGFR3*. In 10% to 20% of the cases, NMIBC can invade the muscular layer and evolve as muscular invasive bladder cancer (MIBC); this evolution is characterized by the loss of the tumor suppressor genes *TP53* and *RB1* [5]. MIBC is the most aggressive BC type, and it is characterized by genomic instability and a high mutational rate. Patients with MIBC tend to present with a 5-year survival rate of 60% when they have a localized tumor, but less than 10% when distant metastases are present [6].

Most NMIBCs are exophytic tumors; however, some lesions present an endophytic growth pattern affecting the lamina propria. Two main subepithelial growth patterns have been described: the endophytic growth pattern (EGP) and the von Brunn’s nest involvement (VBNI). These growth patterns have been associated with high-grade tumors and higher stages of the disease; however, no positive correlations with bladder cancer recurrence have been associated with these growth patterns [7,8].

The molecular analysis of BC has led to the development of different classifications of both NMIBC and MIBC [9,10,11,12]. After an international consensus, MIBC was classified into six different molecular subtypes: Basal/Squamous, Neuroendocrine-like, Stroma-rich, Luminal-papillary, Luminal-unstable, and Luminal-non-specified [6]. Molecular analysis has also identified mutational signatures associated with the mutational processes for carcinogenesis in bladder cancer. One of the most common mutational signatures in bladder cancer is the single base substitution signature 13 (SBS13), which is based on six types of substitutions: C > A, C > G, C > T, T > A, T > C, and T > G, and is related to the cytidine deaminase activity of APOBEC. Other mutational signatures reported for bladder cancer included SBS01 and SBS05 associated with the aging process, SBS02 associated with APOBEC, SBS04 associated with smoking, and SBS22 associated with aristolochic acid exposure [13,14].

The improvements in molecular biology and our understanding of tumorigenesis opened the window to the personalized medicine era for patients with BC [1]. For example, the description of mutational processes affecting bladder cancer tumorigenesis. However, the molecular and genomic data found for BC had been obtained principally from European and European descendant patients; other ethnic groups, such as Hispanics, are underrepresented in these classifications. Here, we aim to identify the somatic single nucleotide variants (SNVs) and copy number variations (CNVs) present in Mexican patients with bladder cancer, exploring their association with clinical-pathological characteristics.

## 2. Results

### 2.1. Clinical-Pathological Characteristics of the BC Patients

We included 37 patients treated at the Instituto Nacional de Cancerología (INCan) between 2012 and 2021. The mean age of patients was 62.46 years (standard deviation [SD]: 11.41 years) with a mean time to follow-up of 39.91 months (SD: 19.44). Men accounted for 70.73% of the patients included in this study. Of the total patients, 59.46% had an educational level of high school or less, and 48.56% had a history of smoking. Nearly half of the patients had a MIBC phenotype (45.95%), whereas 59.46 were found with subepithelial infiltration. Details of the characteristics of the patients are shown in Table 1.

When evaluating the overall survival of the patients, a median value for survival in this cohort was not reached (Figure 1a). When grouping patients according to their muscle invasion phenotype, the patients with MIBC showed a poor overall survival compared to patients with NMIBC (*p*-value = 0.001), with a median survival of 25 months (Figure 1b). In addition, we found a significant difference in the overall survival of the patients when sorting them according to their histological grade (*p*-value = 0.047); however, we did not observe any differences related to overall survival in the Kaplan-Meier curves related to smoking or sex (Appendix A). Additionally, when performing a sensitivity analysis, we found no modification in the influence of muscular invasion on mortality (Hazard Ratio [HR] = 13.11; 95% Confidence Interval [95%CI] = 1.66, 103.60) when adjusting for other clinical characteristics, but a mild decrease when adjusting by histological grade (HR = 6.83, 95%CI = 0.86, 54.02, Appendix A).

### 2.2. Tumor Mutational Burden and Clinical-Pathological Characteristics

Whole exome sequencing (WES) was achieved successfully with an average sequencing depth on the target region of 183.90× and an average coverage of the target region of 99.65%. Samples had a mean of 94.83% Q30 quality in the sequenced bases. FASTQ files were obtained, and further bioinformatics analysis was performed to detect somatic and structural variations. Samples showed a median of 623 somatic variants (ranging from 136 to 2552 mutations per case), including SVNs and InDels (Insertion-or-Deletion). The total number of somatic variants, including driver and passenger mutations, present in the exome region, was used to identify the tumor mutational burden (TMB) of each tumor. The TMB values ranged from 2.27 to 42.53 mutations/Mb. A total of 43% tumors had TMB values greater than 10 mutations/Mb and were classified as having high TMB status (Figure 2). The relationship between TMB and clinical features is shown in Appendix A.

### 2.3. Somatic Variants and Clinical-Pathological Characteristics of the Patients

High-confidence somatic variants were classified as driver and passenger mutations. All the samples had driver somatic variants. The tumor suppressor gene *TP53* was the most frequently affected in our study, which was mutated in 43% of the patients. Other genes which frequently mutated included the gene encoding for the lysine methyltransferase 2C (*KMT2C)* (35%), the fibroblast growth factor receptor 3 (*FGRFR3)* (32%), the cell division cycle protein (*CDC27)* (27%), the lysine demethylase 6A (*KDM6A)* (27%), the SWI/SNF-Related protein *ARID1A* (27%), and lysine methyltransferase 2D *KTM2D* (27%). Missense SNVs were the most common type of somatic variant (Figure 3).

When grouping the patients according to their muscle invasion phenotype, we observed that *TP53* (*p*-value = 0.014, False Discovery Rate [FDR]: 0.54) and *KMT2D* (*p*-value = 0.060, FDR = 0.540) tend to have a high frequency of mutations in patients with MIBC (Figure 4a). All patients with MIBC had a high histologic grade, but not all patients with NMIBC had a low histologic grade. Mutations in *STAG2* were exclusive of patients with NMIBC phenotype and tended to be associated with a low histologic grade (*p*-value = 0.003, FDR = 0.72). Mutations in *FGFR3* also tended to be associated to patients with low histologic grade (*p*-value = 0.005, FDR = 0.729) (Figure 4b). Patients with positive smoking history were mainly associated with a high frequency of mutations of *KMT2D* (*p*-value = 0.015, FDR = 0.514) (Figure 4c). On the other hand, male patients tended to present a high frequency of mutations in *FGFR3* (*p*-value < 0.001, FDR = 0.288), *EP300* (*p*-value = 0.004, FDR = 0.513), and *STAG2* (*p*-value = 0.011, FDR = 0.513) (Figure 4d).

### 2.4. Structural Variants and Clinical-Pathological Characteristics of the Patients

The structural variants were identified for the patients as CNV gains and CNV losses. As shown in Figure 5a, chromosomes 8 and 19 and were affected almost entirely by CNV gains, while chromosomes 9, 10, 17, 21, and 22 were affected mainly by CNV losses. Even though large chromosomic regions were affected by structural variants, our cohort also showed cytobands with a higher frequency of CNVs (Figure 5a). Cytoband chr5q31.3 was the most frequently affected, showing CNV losses in 98% of the samples. Therefore, the protocadherin gamma cluster genes (*PCDHG*), encoded in chr5q31.3, were the most frequently affected genes, with CNV losses in 95% of the patients. *ZMAT2* was also present in this cytoband and was found to be affected in 73% of the patients. Chr1p36.21 was affected completely by CNV losses in 95% of the cases and chr19q13.42 was affected in 82% of the cases by CNV gains and losses. Within chr1p36.21 and ch19q13.42, the genes frequently affected by CNV included: *RPS9* (78%), *LILRB3* (73%), *PRAMEF12* (54%), and *PRAMEF1* (38%). Figure 5b shows an arrangement of the cytobands frequently affected by CNV gains and losses among the patients evaluated and according to their muscle invasive phenotype.

When grouping according to the muscle invasive phenotype of the patients (Figure 6a), those with MIBC tended to show a high frequency of CNV gains in cytobands chr11p15.5 (*p*-value = 0.010, FDR = 0.540) and chr19p13.11-q12 (*p*-value = 0.024, FDR = 0.540). *MUC2* encoded in chr11p15.5 was found in all the patients affected by CNV gains, with log2 values ranging from 0.22 to 2.19. In the same direction, *ZNF429* harbored in chr19p13.11-q12 was found in all the patients affected by CNV gains. CNV gains in chr11p15.5 also tended to be frequent in patients with a high histologic grade, but did not reach the threshold of statistical significance for the raw *p*-value (*p*-value = 0.057, FDR 0.729), whereas CNV losses in chr7q11.23 (*p*-value = 0.005, FDR = 0.729), chr18p11.21 (*p*-value = 0.019, FDR = 0.729), and chr7q35 (*p*-value = 0.021, FDR = 0.729) were frequently affected in patients with low histologic grade (Figure 6b). Regarding the smoking status of the patients, CNV deletions in cytoband chr15q13.1 (*p*-value = 0.046, FDR = 0.513) were found to be as significant in patients with a smoking history (Figure 6c). No significant associations between the sex of the patients and the affected cytobands were found (Figure 6d).

In addition, we found no modification in the influence of muscular invasion on mortality when performing a sensitivity analysis adjusting for the genomic alterations that were found to be associated with muscular invasion (HR = 13.11, 95%CI = 1.66, 103.60). A mild reduction in the influence was observed when adjusting for *TP53* mutation (HR = 8.09, CI95% = 0.95, 68.78); however, no modifications were observed when adjusting for KMT2D mutation, chr11p15.5 amplification, and chr19p13.11-q12 amplification. (Appendix A).

### 2.5. Mutational Processes and Clinical-Pathological Characteristics

The mutational processes implicated in bladder cancer were evaluated through mutational signature analysis, identifying the contribution of single-base substitution signatures (SBS). Signature SBS16, related to liver cancer and defective nucleotide excision repair pathway, was the most frequent mutational process, with a contribution greater than 0.2 and found in 89% of the patients, followed by SBS05 present in 51% and SBS13 in 19% of the patients (Figure 7a). When evaluating the association of the mutational processes, we observed that patients with smoking history were associated with greater contribution values of SBS05 (Appendix A). High TMB status was also associated with SBS02 and SBS13, whereas a low TMB status was associated with SBS01 and SBS05 (Figure 7b).

## 3. Discussion

To our best knowledge, this is the first report of the somatic mutational landscape of bladder cancer in Mexican patients. Using WES, we identified a differential mutational pattern between patients with MIBC and NMIBC. Patients with MIBC were associated with a higher frequency of mutations in *TP53* and *KMT2D* and CNV gains in chr11q15.5 and chr19p13.11-q12. In contrast, mutations in *STAG2* and CNV deletions in chr7q11.23 were associated with patients with NMIBC and a low histologic grade.

The patients included in our study showed similar characteristics to other previously published Mexican bladder cancer studies. A descriptive epidemiological study of 20 years of experience with bladder cancer in the INCan identified a median age at diagnosis of 54 years with a predominance of male patients and smoking as the main risk factor. A high rate of patients were diagnosed in late disease stages, with 70% of the cases with metastasis [15]. Another study performed in a general hospital in Mexico reported a median age at diagnosis of 62.5 years with a predominance of male patients. Within this study, a prevalence of 36% was reported for patients with MIBC [16].

Overall, non-Hispanic whites are the ethnic group with the highest incidence of bladder cancer, followed by Blacks and Hispanics [8]. These differences are also seen to be associated with the overall survival of the patients, with non-Hispanic whites as the ethnic group with the best rates [8]. Differences in the tumor biology and somatic mutations could be influenced by the ethnicity of the patients; a higher frequency of *TP53*, *ARID1A*, *ERBB3,* and *CDKN1A* mutations have been reported for white patients compared to non-white patients [17]. *TP53* was the gene most frequently mutated in our cohort. Similar to reports in other populations [18], we identified an association between TP53 mutation and muscular invasion. In addition, mutations in *TP53* have been associated with poor overall survival [18]. Within the multivariate analysis performed in this study, we observed a positive influence of *TP53* on the mortality of the patients, suggesting that mutations in *TP53* affect Mexican patients in the same way that affects other populations.

This group of patients evaluated included a similar proportion of patients with MIBC and NMIBC, with a greater proportion of cases with high histologic grades. We identified *KMT2D* as a gene frequently mutated in patients with MIBC and high histologic grade. *KMT2D* encodes the lysine methyltransferase 2D; the loss of this gene has been associated with abnormal epigenetic reprogramming of different molecular pathways [19]. *KMT2D* is mutated in different types of cancer, including bladder cancer [20]. In bladder cancer cell lines, *KMT2D* overexpression was associated with tumor suppressor effects, as it promoted the expression of *PTEN* and *TP53*, as well as the repression of *STAG2* [21]. In lung cancer, it has been observed that *KMT2D* mutations increase the glycolysis of tumor cells. For this reason, it has been proposed to study the pharmacological inhibition of glycolysis in tumors deficient in *KMT2D* [22]. The expression of *KMT2D* has also been proposed as a prognostic biomarker for bladder cancer in European patients, observing a trend towards greater survival in patients with higher expression of this protein [23]. However, it is unknown whether this effect was also present in Mexican patients.

We also observed that patients with MIBC frequently showed copy number gains of the chr11p15.5 cytoband. *MUC2* was seen to be affected in all patients with these gains. This protein has been seen to contribute to the development of colorectal cancer [24]. Similarly, there is evidence suggesting that the expression of *MUC2* is associated with invasion and metastasis in various malignant tumors, including gastric cancer, prostate cancer, and colorectal cancer [25]. In the case of bladder cancer, the presence of *MUC2* has been associated with the non-invasive proliferation of tumors or with a favorable outcome for patients [26]. It has also been seen that the expression of *MUC2* is observed more frequently in low-grade urothelial carcinomas and correlates with a low pathological stage [26].

Within our cohort, nearly half of the patients were associated with TMB values greater than 10 mutations/Mb. TMB has been used as a biomarker for the selection of different immune checkpoint inhibitors presenting a wider range of therapeutic strategies for the treatment of bladder cancer patients [27,28]. To our best knowledge, there are no studies exposing the clinical uses of TMB in Mexican patients with bladder cancer. We found no associations between High-TMB and clinical characteristics, which could mean a greater number of patients that could benefit from TMB as a biomarker for treatment selection.

We are aware that the limited number of patients included in this study could hinder the statistical power of the analysis. Nonetheless, this study provides new information about the clinical and pathological characteristics of Mexican patients regarding bladder cancer, which could guide us in the design of further studies that feature a larger cohort to test some of the identified variants as biomarkers of prognosis for patients treated.

## 4. Material and Methods

### 4.1. Population

A total of 37 patients treated between 2012 and 2021 at the Instituto Nacional de Cancerología (INCan), located in Mexico City, were selected randomly for this study from the total of patients that met the inclusion criteria. Inclusion criteria involved patients with formalin-fixed paraffin-embedded (FFPE) tumor tissue blocks stored at the pathology department of the INCan, who were older than 18 years old and had a positive diagnosis of urothelial bladder cancer. FFPE samples were collected by transurethral resection of the bladder tumor (TURBT) before any kind of treatment. Patients with a history of other cancer or any type of treatment prior to TURBT were not included in this study. Tumor samples were selected from formalin-fixed paraffin-embedded tissue (FFPE) blocks for an experienced pathologist. Selected tumor samples had at least 70% of tumor cellularity. This project was approved by the institutional ethics board (017/034/IBI) and the institutional ethics in research committee (CEI/1175/17) of the INCan. Due to the minimal risk and the nature of this study for the patients and following the guidelines of the institutional ethics board, the informed consent was waived.

### 4.2. Clinical Data Collection

Clinical data, including age at diagnosis, body mass index (BMI), sex, education level, smoking status, symptoms at diagnosis, muscular invasion, subepithelial invasion, and histological grade, were collected from the electronic clinical records of each patient. The follow-up time was calculated from the clinical records as the period between the date of diagnosis and the date of death or loss of follow-up.

### 4.3. DNA Extraction and Quality Control

Samples were homogenized using QIAshredder (QIAGEN, 79654), and DNA was extracted using the QIAamp^®^ DNA FFPE Tissue Kit (QIAGEN, 56404), following the protocol recommendations. Purity of DNA was evaluated with a Thermo Fisher Scientific NanoDrop 2000. Quantity and DNA fragmentation status were evaluated with the Agilent 2200 TapeStation System using the genomic DNA ScreenTape assay (Agilent, 5067–5365) to obtain a DNA integrity number [29]. Samples with DIN from 6 to 10 and a minimum concentration of 20 ng/µL were selected for WES.

### 4.4. Library Preparation, Hybridation Capture, and WES

Library preparation, hybridization capture, and WES procedures were performed by the New York Genome Center. TruSeq DNA PCR-Free libraries were prepared from FFPE tissues using 1µg of input DNA according to the manufacturer’s instructions (Ilumina, San Diego, CA, USA). Sequencing was performed on HiSeq2500 (Ilumina, San Diego, CA, USA).

### 4.5. Bioinformatics Pipeline

Sequencing reads for the tumor samples were first trimmed for adapters using TrimGalore (v0.4.0). The trimmed reads were then aligned to the reference genome using BWA-MEM (v0.7.15) [30], GATK (v4.1.0) [31] FixMateInformation was run to verify and fix mate-pair information, followed by Novosort (v1.03.01) markDuplicates to merge individual lane BAM files into a single BAM file per sample. Duplicates were then sorted and marked, and GATK’s base quality score recalibration (BQSR) was performed to obtain a coordinated sorted BAM file for each sample.

A matched normal sample was not available. In its place, we used the HapMap sample NA12878, which was prepared and sequenced using the same protocol as the tumor sample. This normal sample was used to remove some of the false positives that were due to library preparation and sequencing (that would manifest in the same way in the tumor and NA12878), as well as some germline variants that were common to the tumor sample and NA12878.

The tumor and normal BAM files were processed using (GATK v4.0.5.1) [31], Strelka2 (v2.9.3) [32], and Lancet (v1.0.7) [33] for calling SNVs and short Indels. SvABA (v0.2.1) [34] for calling Indels. FACETS (v0.5.5) [35] for calling CNVs. High-confidence variants identified by at least two variant callers and variants with a variant allele frequency equal to or in between 0.1 and 0.45 were selected for subsequent analysis.

SNVs and Indels were annotated with Ensembl, as well as databases such as COSMIC (v86) [36], 1000Genomes (Phase3) [37], ClinVar (201706) [38], PolyPhen (v2.2.2) [39], SIFT (v5.2.2) [40], FATHMM (v2.1) [41], gnomAD (r2.0.1) [42], and dbSNP (v150) [43] using Variant Effect Predictor (v93.2) [44]. Synonymous mutations and mutations annotated in non-coding regions were filtered out.

For CNVs, segments with log2 > 0.2 were categorized as amplifications, and segments with log2 < −0.235 were categorized as deletions (corresponding to a single copy change at 30% purity in a diploid genome, or a 15% Variant Allele Fraction). CNVs of a size less than 20 Mb were denoted as focal and the rest were considered large-scale. Only focal CNV were selected for subsequent analysis. We used Bedtools [45] for annotating CNVs. All predicted CNVs were annotated with germline variants by overlapping with known variants in 1000 Genomes and Database of Genomic Variants (DGV) [46].

### 4.6. Mutational Signature Analysis

Mutational signatures analysis of single-base substitutions (SBS) was performed to identify the contribution of the different mutational processes to carcinogenesis according to COSMIC database [47,48]. Mutational signatures were calculated for each sample using the R package deconstuctSigs (v2) [49]. COSMIC signatures were set as reference signatures, and the count method was set as default. Mutational signature weight ≥0.2 was considered a positive contribution to the mutational process. The “tri.counts.method” parameter in deconstructSigs was set to “exome2genome” and a custom exome trinucleotide counts file based on the target interval was provided.

### 4.7. Statistical Analysis

Associations between clinical characteristics such as muscular invasion, histologic grade smoking history, and sex were assessed with the presence of somatic variations and CNVs. Patients were grouped according to their clinical characteristics, then the frequency of the variants between the groups was compared using Fisher’s exact test. *p*-values were adjusted using the Benjamini–Hochberg procedure. The Kaplan-Meier curve was analyses by Log rank test. The mutational signatures contribution was compared with the clinical characteristics of the patients using the Mann-Whitney U test. Statistical significance was set at *p*-value < 0.05. All statistical analyses were performed using R software (https://cran.r-project.org, accessed on 23 March 2022).

## 5. Conclusions

This is the first work in a Mexican population in which the mutational panorama of BC is characterized for both MIBC and NMIBC. Patients with MIBC showed a higher frequency of mutations in *TP53* and *KMT2D*, gains in chr11q15.5 and chr19p13.11-q12, and losses in chr7q11.23. *STAG2* mutations and CNV deletions at chr1q11.23 were frequently found in patients with NMIBC and low histologic grade. These genomic changes may open new research lines toward their specific detection at diagnosis in prospective studies.

## Figures and Tables

**Figure 1 ijms-24-01092-f001:**
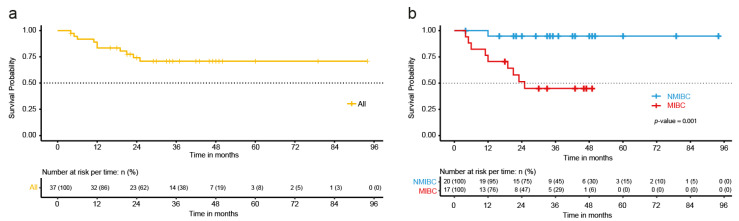
Overall survival of the patients with bladder cancer treated at the INCan (N = 37) between 2012 and 2021; (**a**) Kaplan-Meier curve depicting the overall survival of all the patients included in the study; (**b**) Kaplan-Meier curve showing the overall survival differences between the patients with non-muscle invasive bladder (NMIBC) cancer and muscle-invasive bladder cancer (MIBC); The *p*-value was calculated using the Log Rank test.

**Figure 2 ijms-24-01092-f002:**
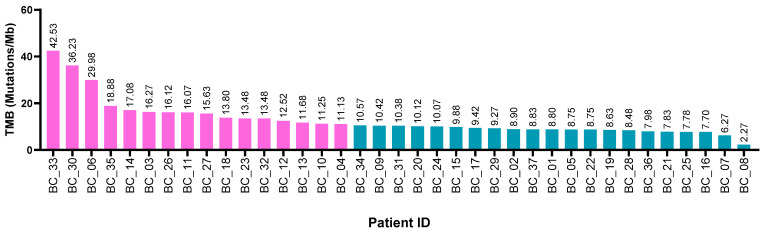
Tumor mutational burden (TMB) patients with bladder cancer treated at the INCan (N = 37) between 2012 and 2021. Bar plot depicts TMB distribution among all the patients. Pink bars represents patients with high TMB, and blue bars represents low TMB.

**Figure 3 ijms-24-01092-f003:**
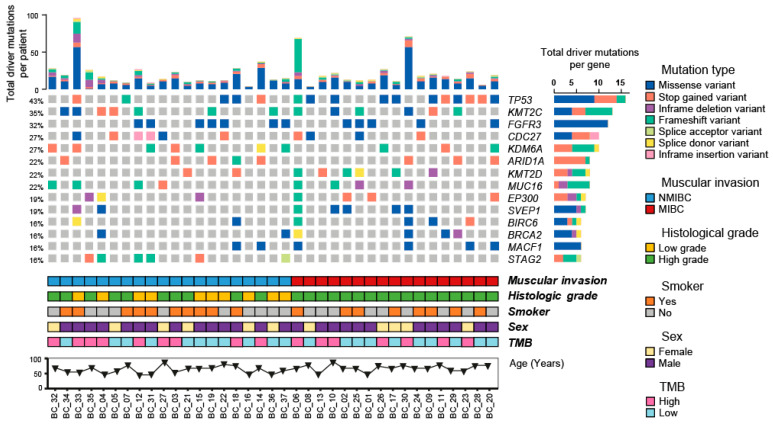
Driver somatic variants and clinical characteristics of patients with bladder cancer treated at INCan between 2012 and 2021 (N = 37). Oncoprint, sorted by muscular invasion (Non-muscle invasive bladder cancer [NMIBC], n = 20 and muscle invasive bladder cancer [MIBC], n = 17), depicting the genes affected by driver somatic variants (single nucleotide variations, small insertions, and small deletions) in 16% or more of the samples. The variants are represented according to the mutation type, each described on the right color panel. The upper bar plot represents the number of driver somatic variants per patient, and the right bar plot represents the number of driver somatic variants per gene. The lower paneer represents the muscular invasion, histologic grade, smoking history, and sex of the patients.

**Figure 4 ijms-24-01092-f004:**
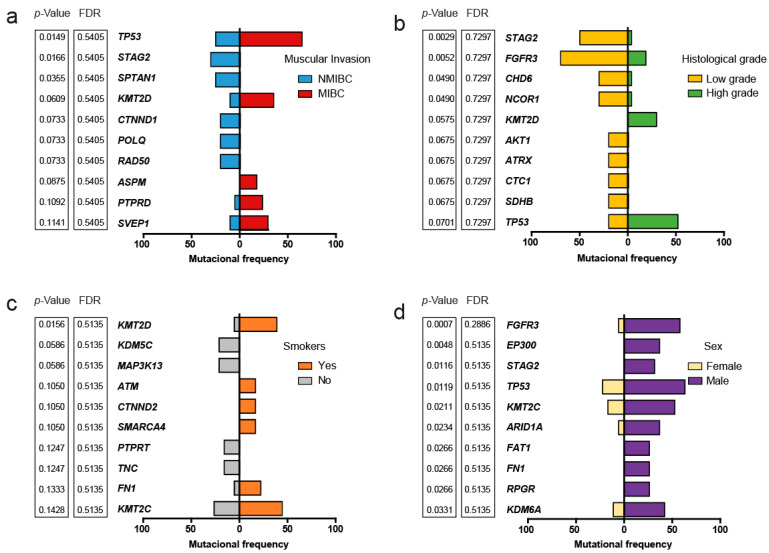
Muscle invasion, histologic grade, smoking status, sex, and the frequency of somatic variants. The 10 most affected genes were ordered in ascending order according to smallest *p*-value on the top, obtained with the Fisher’s exact test; (**a**) Comparison of somatic variant frequency according to the muscular invasion phenotype; (**b**) Comparison of somatic variant frequency according to the histologic grade; (**c**) Comparison of somatic variant frequency according to the smoking status; (**d**) Comparison of somatic variant frequency according to the sex of the patients.

**Figure 5 ijms-24-01092-f005:**
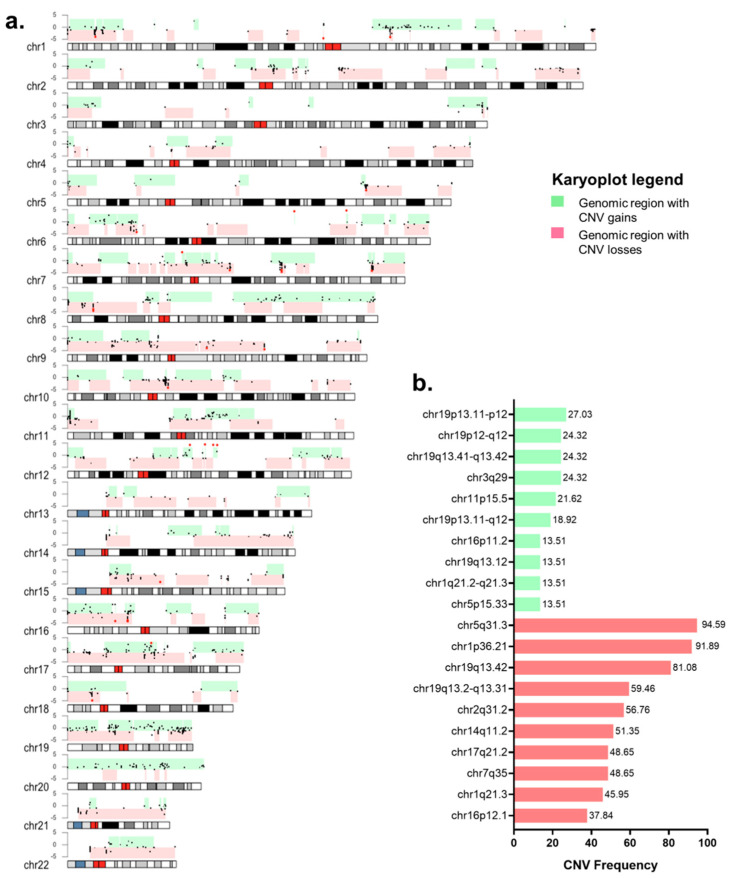
Structural variants present in Mexican patients with bladder cancer treated at the INCan (N = 37). (**a**) Karyoplot depicting the regions on each chromosome affected by copy number variations (CNV). The pink bars depict the genomic regions with CNV losses, and the green bars indicate the genomic regions with CNV gains on each chromosome present by one or more patients. The black dots within the pink and green bars indicate the start position of the CNV event per patient. The red dot indicates CNV events with a log2 greater than 3 or lower than ₋ 3 per patient; (**b**) Bar plot presenting the ten cytobands more frequently affected by CNV gains and losses in the cohort. Green bars indicate CNV gains, and pink bars indicate CNV losses.

**Figure 6 ijms-24-01092-f006:**
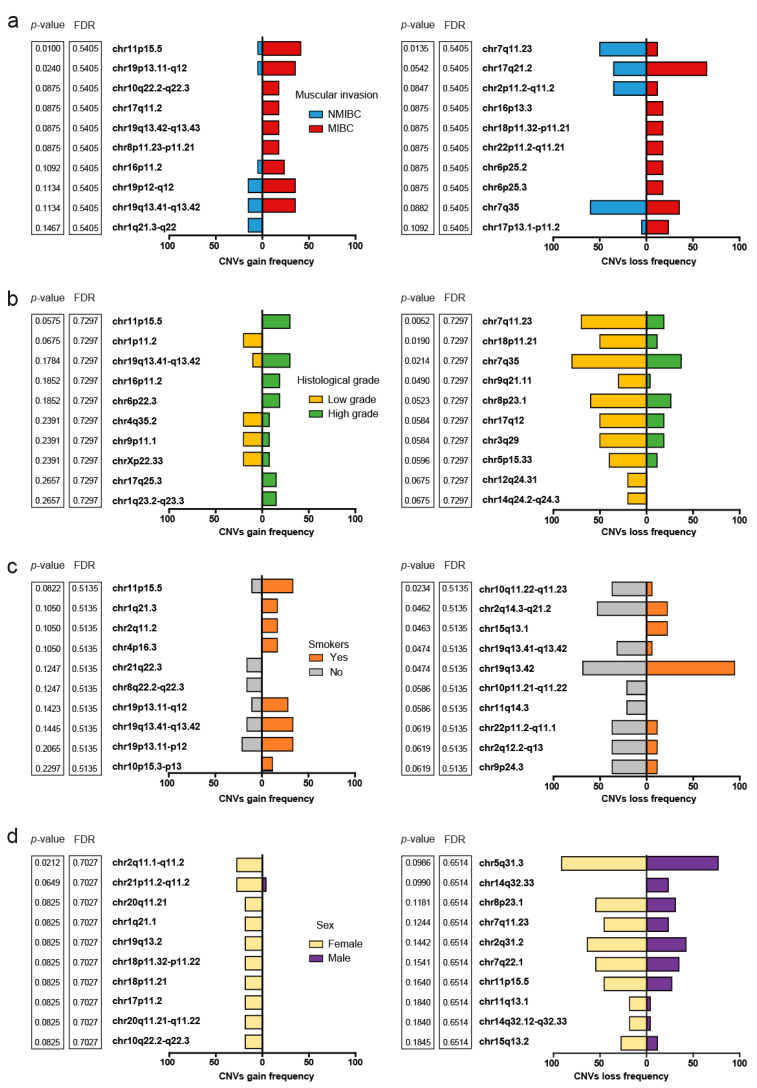
Muscle invasion, histologic grade, smoking, gender, and frequency of cytobands affected by CNV. The 10 most affected cytobands are ordered in ascending order from the smallest *p*-value according to Fisher’s exact test on top: (**a**) Comparison of the frequency of gains and losses in copy number according to the muscle invasion phenotype; (**b**) Comparison of the frequency of copy number gains and losses according to histological grade; (**c**) Comparison of the frequency of copy number gains and losses according to smoking; (**d**) Comparison of the frequency of gains and losses in the number of copies according to the sex of the patients.

**Figure 7 ijms-24-01092-f007:**
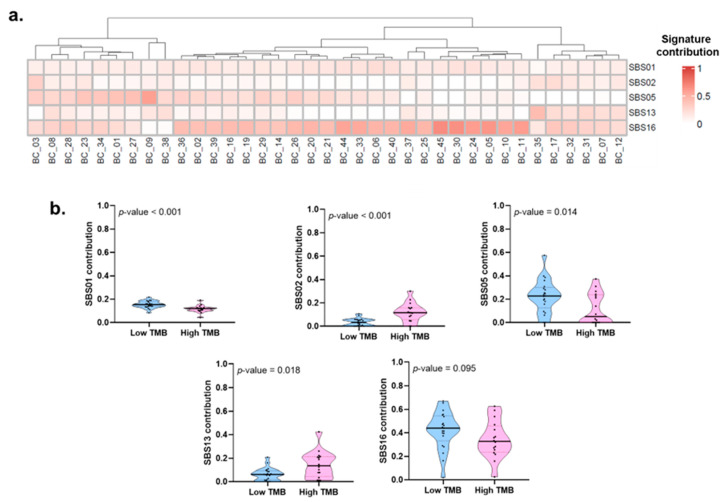
Mutational processes and their association with TMB status of patients with bladder cancer treated at the INCan (N = 37) between 2012 and 2021: (**a**) Heatmap showing the contribution of the mutational signatures to the carcinogenic process. Only single-base substitution signatures (SBS) with a contribution more significant than cero were depicted. The gradient red color indicates the grade of the contribution of each signature; (**b**) Violin plot depicting the association between the tumoral mutational burden (TMB) status of the patients and SBS01, SBS02, SBS05, SBS13, and SBS16. We did not find any association between TMB and clinical features (Appendix A).

**Table 1 ijms-24-01092-t001:** Demographic characteristics of patients with bladder cancer treated at the INCan between 2012 and 2021 (N = 37).

Variable	Mean/n	SD/%
Age (Years)	62.49	11.41
BMI (kg/m^2^)	26.75	3.99
Time to follow-up (Months)	39.91	19.44
Sex		
Women	11	29.73%
Men	26	70.73%
Education level		
High school or less	22	59.46%
College or vocational school	5	13.51%
Grad school or higher	10	27.03%
Smoking	18	48.65%
Symptoms at diagnosis		
Asymptomatic	1	2.70%
Symptomatic ^1^	2	5.41%
Hematuria	34	91.89%
Muscle invasion disease		
Yes	17	45.95%
Subepithelial infiltration		
Yes	22	59.46%
Histological grade		
Low	10	27.03%
High	27	72.97%
Recurrence	11	29.73%

SD = standard deviation; ^1^ Symptomatic = Irritative symptoms during urination (dysuria or burning sensation); BMI: Body Mass Index.

## Data Availability

The data presented in this study are available upon request to the corresponding author if you want to partner with or contribute to the project. The data are not publicly available due to Institutional Review Board policy.

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
