# Peer review of "Mutational Landscape of Bladder Cancer in Mexican Patients: KMT2D Mutations and chr11q15.5 Amplifications Are Associated with Muscle Invasion"

_ijms, 2023, doi:10.3390/ijms24021092_

Round 1

Reviewer 1 Report

This interesting manuscript describes mutations and copy number variations (CNVs) in a small cohort of bladder tumours from Mexican patients. The work is important because, as the authors correctly state, Hispanic populations are poorly represented in current cancer genetic/genomic databases. Overall, the study has been appropriately designed, and presentation of the work is clear, concise, and generally easy to follow. I feel that the study is suitable for publication, however there are a number of details that should be added to the manuscript to increase the usefulness of the large amount of data that has been generated.

1. A total of 37 patients is a low number for this type of study. Does this represent all bladder cancer patients treated at this hospital, or is it a selection of patients? If it does represent a subset of the patients, how were these selected and what are the potential impacts of this bias.

2. Minor errors: Two of the supplementary figures have been labelled as “Supplementary S2”. This should be corrected. “Codifies for” (line 169) should be “encodes”.

3. When genetic/genomic screens are performed, it is usual for the raw results to be either included as supplementary information or to be uploaded into one of several freely available public repositories. By doing this, other researchers can either mine the information or add/compare it to other datasets or to their own results. I would encourage the authors to consider doing this.

4. The labelling for Figure 5a does not seem to match the colours or indications in the figure. Could the authors please re-check the figure legend?

5. There is insufficient detail regarding each of the types of analysis performed on the sequencing data. A major target audience for this type of manuscript would be bladder cancer researchers, who would not necessarily have an in-depth knowledge of the different types of genomic analysis that have been applied to the raw results. Addition of a sentence or 2 and a couple of relevant references (apart from an original reference describing development of the analysis platform) would broaden the readership and the application of results from this study. This comment is particularly relevant for the description of single-base substitution signatures – there is insufficient information even for those who are familiar with these signatures.

6. I don’t feel that there is enough of an analysis of results in the Discussion section of the manuscript, and because the results descriptions are too limited, new information is being introduced in the Discussion section. In addition, the authors have not discussed whether they found any differences in the mutation or CNV profiles of bladder cancers in their population in comparison to the published databases where data are predominantly derived from people of European descent. I would recommend that the Discussion section is expanded to include a more comprehensive analysis of the results achieved and the implications of those results in relation to both current knowledge of bladder cancer genetics/genomics, and bladder cancer treatment.

Reviewer 2 Report

What is missing in the manuscript „Mutational landscape of bladder cancer in Mexican patients: KMT2D mutations and chr11q15.5 amplifications are associated with muscle invasion“ is a good multivariate analysis.

Cancer is a multifactorial disease and the genetic background of an extirpated tumor highly depends on the host's genetics, exposure to various stressors and the time during which tumor evolves.

I presume that the tumors were chemotherapy naive, but that was not stated. Also, the term „SBS“ and the biological background for its occurrence as an phenomenon must be described in the Introduction.

When taking a good look at Fig 5, I see no chromosome 8 gains on the „b“ part of the figure. Also, I see no gain of chr. 20, which is well-visible on the „a“ part of the figure. A valid explanation is needed. I understand that „b“ shows specific chromosomal location, only, but that needs to be well-connected and explained to what is presented on the „a“.

The Discussion must be significantly improved. The authors accentuate the origin of patients (Mexico). That is important, I agree. But the uniqueness of  a population must be considered with respect to results/data obtained in other populations.

Also, the authors need to discuss the mutational status of frequently mutated genes- KMT2C, FGRFR3, CDC27, KDM6A, ARID1A,and KTM2D- with respect to other studies performed on BC. One would want to see  the full name of genes and corresponding proteins. The basic function of proteins coded by these genes must be described.

Minor items:

In the Introduction, the meaning and significance of the term „subepithelial infiltration“ needs to be given. Numbers for the two basic subgroups must be given (NMIBC=20; MIBC=17), as presented on Fig. 3. Fig 3: the graph showing age should be improved. There is no sense in showing 100 years and losing resolution by doing so. I see no reason for listing genes with a p-value higher than 0.05 (also presented on other figures). For example, with respect to gender, only a gain of chr2q11.1-11.2 seems to be significant. From a technical point of view, numbers in the text must be identical to those on figures. Figure 5: I am not sure how to present percentages, but, currently, the frequency of specific CNV is not well-presented.

The word „patient“ should be replaced with the word „tumor“, whenever appropriate, as is in the title of  subsection 2.3.

Is it to be expected that there is no statistically significant difference between high- and low grade tumors with respect to OS? Please, discuss.

As a general recommendation, I would suggest that the authors present what is really important and discuss it thoroughly, instead of presenting all that was obtained without proper discussion.

Thank you.

Round 2

Reviewer 1 Report

The authors have answered reviewers’ comments and I feel that the manuscript is suitable for publication in its present form. There are minor grammatical errors that require correction. Other minor errors are listed below.

1. Line 78: The second half of this sentence is missing.

2. Line 153: ‘paneer’ should be ‘panel’

Reviewer 2 Report

Dear authors,

Thank you for accepting my suggestions.